# Structure Design of Quadrilateral Overlapped Wireless Power Transmission Coupling Coil

**DOI:** 10.3390/s22165955

**Published:** 2022-08-09

**Authors:** Xiaotian Wang, Changli Yu, Yuteng Wu, Jingang Wang

**Affiliations:** State Key Laboratory of Power Transmission Equipment & System Security and New Technology, School of Electrical Engineering, Chongqing University, Chongqing 400044, China

**Keywords:** wireless power transmission system, new coil structure, anti-offset, transmission efficiency

## Abstract

The application of wireless power transmission technology is becoming more and more extensive. However, in practical applications, the problem of reducing the energy transmission efficiency caused by the offset of the coupling coil needs to be solved urgently. Changing the coil structure is a widely adopted method to deal with this problem. Based on the characteristics of the existing magnetic coupling resonant wireless power transmission system and the principle of the anti-offset coil, this paper innovatively designs a new type of quadrilateral overlapping wireless power transfer coupling coil, which has a strong anti-offset capability. The new type of coil model was built in the simulation and experiment, and the relevant parameters were measured. Experimental results verify that the proposed coil structure has an excellent anti-offset capability.

## 1. Introduction

### 1.1. Background

Wireless power transmission (WPT) is a popular technology for research and application today. Compared with traditional wired power transmission technology, wireless energy transmission technology has the advantages of flexibility, convenience, electrical isolation, strong environmental adaptability, and easy maintenance. In recent years, radio transmission technology has been gradually applied in various fields and has produced far more than the expected results. It brings many advantages in specific domains such as electric vehicles [1,2,3], implantable medical devices [4,5,6,7], consumer electronics [8,9,10,11,12,13], industrial robots [14,15,16], underwater electrical equipment [17,18,19,20], and so on, which have aroused great interest in industry and academia.

At present, the wireless power charging technology is mainly divided into three types: the microwave type, electromagnetic induction type, and magnetic coupling resonance type. This paper focuses on magnetically coupled resonant wireless power transmission (MCRWPT), which is based on two or more coils with the same resonant frequency and uses the principle of magnetic field resonance to achieve efficient energy transmission. This technology has the advantages of non-radiative energy, being not easily affected by non-magnetic obstacles, high transmission efficiency, and large output power [21] and has broad application prospects in electric vehicles and other industries.

Magnetically coupled resonant wireless power charging technology has developed rapidly, but there are still many problems to be solved. The transmission efficiency of the system is determined by the relative positions of the transmission and reception coils, but coil offset is unavoidable in practical applications, especially in the equipment of single-transmit and multi-receive wireless energy transmission systems. It is difficult to share the highest point of transmission efficiency when there are multiple receivers, which leads to a decrease in the efficiency of system energy transfer. Thus, the anti-offset of the system is an important problem [22], and one of the usual solutions is to improve the coil structure. The new coil structure studied in this paper will help to improve the problem of reducing the transmission energy when the magnetic coupling coil is offset. It also has great potential to be applied to the wireless power transmission system of a single transmitter and multiple receivers.

### 1.2. Research Status at Home and Abroad

Magnetic coupling resonance technology was first proposed in 2006 by Professor Marin Soljacic of the Massachusetts Institute of Technology at the American Physics Industrial Physics Forum. The following year, two copper coils with the same structural parameters were used to light up the Columbia World Expo. A 60 W light bulb was tested at a distance of 2.13 m, and the system transmission efficiency was about 40%, which verified the feasibility of the magnetic coupling resonance technology [23].

In 2008, Joshua R. Smith’s research group at Intel’s Seattle Laboratory used magnetic coupling resonance technology to achieve wireless charging of notebooks, PDAs, and other appliances with 60 W output power and 75% transmission efficiency when the transmission distance was 1m. In 2008, Professor Sun Mingus from the University of Pittsburgh first applied magnetic resonance technology to the power supply of implanted electronic devices in animals. The operating frequency is about 7 MHz When the axial distance between the two coils is 20 cm, it can output watt-level power, and the transmission efficiency reaches above 50. In 2013, the team of Professor John T. Boys of Auckland University improved the topology of the coupler, which can improve the coupling coefficient of the coupler [24,25].

China started a little later, researching magnetic coupling resonance technology, mainly on the research and verification of transmission mechanism and characteristics, and related research is also mainly concentrated in university research institutes. The team of Professor Zhu Chunbo of the Harbin Institute of Technology produced an LC series resonator with a radius of 250 mm. When the transmission distance is 0.7 m and the transmission power is 23 W [26,27,28,29,30]. Professor Huang Xueliang of Southeast University developed the first domestic wireless charging device in 2013. Professor Yang Qingxin from Tianjin University of Technology led a team to build a wireless charging project based on a wind-solar hybrid DC microgrid, reaching a charging power of 6 KW [31].

It can be seen that the magnetic coupling resonant wireless energy transfer technology is still popular, and increasing the performance of the system energy transmission efficiency is also a research highlight, including improving the coil structure to reduce the impact of offset on the wireless charging system.

## 2. Theoretical Analysis

### 2.1. Principle of Wireless Power Transmission System

The basic schematic diagram of wireless charging technology is shown in Figure 1. The main modules include the power electronic converter module, compensation network module, coil module, high-frequency rectification filter module, power supply, load, and other parts.

Power electronic converters are used to convert 50/60 Hz alternating current into high-frequency alternating currents to enhance the effect of electromagnetic induction. The compensation network is used to improve the quality of the alternating current signal, transmission efficiency, and system stability. Let *M* is the mutual inductance coefficient, *I* be current, and the coil-induced electromotive force is shown in (1).
(1)$$\left\{ \begin{array}{l} \varepsilon_{12}=-\frac{d\varphi_{12}}{dt}=-M\frac{dI_{1}}{dt}\\ \varepsilon_{21}=-\frac{d\varphi_{21}}{dt}=-M\frac{dI_{2}}{dt} \end{array}\right.$$

The existence of the coil-induced electromotive force means the induced current is generated in the receiving end loop. The current is adjusted to provide electrical energy to the receiving end impedance by the rectifier circuit. To ensure the high transmission efficiency of the system, it is necessary to set the capacitance parameters so that making the transmitting and receiving circuits (composed of coils and capacitors) are in a resonant state.

In practical applications, the relative positions of the receiving coil and the transmitting coil of the wireless energy transmission system are likely to change within a certain range, resulting in a change in the coupling coefficient. The offset has a large impact on the transmission efficiency and leads to system instability.

### 2.2. Relationship between System Transmission Efficiency and Coil Offset

The magnetic coupling resonant circuit model of the wireless power transfer system is shown in Figure 2.

*AC* is the *AC* power supply. *R*_1_ is the sum of the internal resistance of the *AC* power supply, the radiation resistance of the transmitting coil, and the ohmic loss resistance. *C*_1_ and *C*_2_ are the compensation capacitors of the transmitter and receiver, respectively. *L*_1_ and *L*_2_ are the transmitters. *R_L_* is the load resistance of the receiving end. *I*_1_ and *I*_2_ are the total currents of the two loops. According to Kirchhoff’s voltage (KVL) law, the equilibrium Equation (2) is obtained.
(2)$$\left\{ \begin{array}{l} Us=\left[R_{1}+j\left(\omega L_{1}+\frac{1}{\omega C_{1}}\right)\right]I_{1}+j\omega MI_{2}\\ 0=\left[R_{2}+j\left(\omega L_{2}+\frac{1}{\omega C_{2}}\right)\right]I_{2}+j\omega MI_{1} \end{array}\right.$$

When the two coils resonate, and there is ωL+1/ωC=0, bring it into Equation (2) to obtain *I*_1_ and *I_2_*, as shown in Equation (3).
(3)$$\left\{ \begin{array}{l} I_{1}=\frac{R_{1}+R_{2}}{{\left(\omega M\right)}^{2}+R_{1}R_{2}+R_{1}R_{l}}\\ I_{2}=\frac{j\omega MUs}{{\left(\omega M\right)}^{2}+R_{1}R_{2}+R_{1}R_{l}} \end{array}\right.$$

In summary [28], the system transmission efficiency Formula (4) is derived.
(4)$$\eta =\frac{I_{2}{}^{2}R_{l}}{UsI_{1}}=\frac{{\left(\omega M\right)}^{2}R_{l}}{\left[\omega M^{2}+R_{1}R_{2}+R_{1}R_{l}\right](R_{2}+R_{l})}$$

It can be seen from the above power formula that *η* has a positive correlation with the mutual inductance *M*. For a fixed system, the component parameters and loads of the system are unchanged. When the coil is offset, the change of mutual inductance causes the system’s wireless transmission efficiency to change. A schematic diagram of the coil offset is shown in Figure 3.

In Figure 3, the ellipse is simplified to replace the coil, and there are two common offset forms: radial offset and angular offset. $N_{a}$ and $N_{b}$ are the number of turns of the two coils, $r_{ab}$. is the geometric distance between the transmitting and receiving coil micro-elements, $\mu_{0}$ is the vacuum permeability, and α is the angle offset. The mathematical expression between the mutual inductance between coils and the offset parameter is shown in Formula (5) [32].
(5)$$M=\frac{N_{a}N_{b}r_{a}r_{b}\mu_{0}}{4\pi }\frac{\oint d\phi \oint \left[\left(\sin\theta \sin\alpha +\cos\theta \cos\alpha \right)\right]}{r_{ab}}d\theta$$

Changes in coil position and offset affect the mutual inductance of the coils, resulting in changes in transmission efficiency. The change in mutual inductance becomes more pronounced with further distance from the ideal alignment state.

## 3. Structural Design

### 3.1. Analysis Method of Coil Anti-Offset Ability

During the charging process, an offset inevitably occurs between the two coils in the system coupling structure, thereby affecting the mutual inductance between the two coils and reducing the transmission efficiency of the system. This has become a key factor restricting the wide application of wireless charging technology [31]. The coupling mutual inductance between the coils is an important factor affecting the energy transmission performance of the system [33]. The change of the mutual inductance means that the magnetic flux passing through the receiving coil has changed because the offset occurs, and the magnetic field distribution in the coupling area directly determines the magnetic flux through the pickup coil. Therefore, the smaller the change in the magnetic flux during offset, the better the anti-offset performance of the coupling mechanism with more uniform magnetic field distribution in the coupling area. In conclusion, it is very necessary to study the magnetic field distribution characteristics of the coil structure [34].

Optimizing the design of the coil structure can improve the anti-offset characteristics of the coupled coil [34] and the spatial magnetic field distribution generated by the coil. There are usually two ideas for improving the coupling structure [35]. The first is to concentrate the magnetic field generated by the coil even if the coil is offset, which does not lose most of the magnetic flux. The second is to make the magnetic field generated by the coil uniform, and the mutual inductance and coupling coefficient can be maintained even if an offset occurs.

The efficiency of the magnetic coupling mechanism mainly depends on several factors, such as resonant frequency, the size and design of the coupling structure of the transmitting coil and the receiving coil, and the distance between the magnetic coupling structures [31]. The ANSYS simulation platform is used to build the simulation model, and the mutual inductance between the magnetic coupling coil structure with different turns and structures is calculated using the ANSYS HFSS simulation software [36].

### 3.2. Simulation of the Circular Wound Coil

In ANSYS HFSS, the influence of the position change (offset) of the simplest circular-wound coil on the mutual inductance coefficient is analyzed and the law of coil offset is summarized. Figure 4 shows the measured data when the coils are co-displaced in radial, axial, angular, and radial and axial directions.

It can be seen from Figure 4 that the mutual inductance is the largest when the coil is displaced in the radial direction. The mutual inductance coefficient gradually decreases when migrating toward both sides, and the mutual inductance coefficient decays slowly at the beginning. When offset along the axial direction, the mutual inductance coefficient between coils decreases with the increase of offset distance, and the attenuation amplitude is the largest in the initial stage. Therefore, the coil needs to be kept in the right direction and at an appropriate distance, which promises systems have a high transmission efficiency and the influence of a small axial offset is much greater than that of the radial direction.

The curve of angular migration is similar to that of radial migration. When the offset amplitude is small, the change of mutual inductance is small, and the subsequent mutual inductance coefficient decays faster. In practice, the offset of the coil is often in the order of millimeters, and the axial offset, radial offset, and large angular offset can be easily avoided. Thus, anti-offset research focuses more on reducing the energy caused by small radial offsets. The anti-migration capability mentioned later also refers to the resistance to the influence of radial offset.

### 3.3. Design of the New Quadrilateral Overlapped Wireless Power Transmission Coupling Coil

#### 3.3.1. Structural Derivation and Magnetic Field Simulation of the New Coil

The magnetic field of an anti-offset cross helical coil structure is uniformly distributed along a diagonal line of the square magnetic core, and the magnitude of the diagonal magnetic field is larger than that of other regions, which have better anti-offset ability [34]. Use ANSYS to model and simulate the cross-coupled coil, and the corresponding model and magnetic field distribution diagram are shown in Figure 5 and Figure 6 below.

From the above Figure 6, the magnetic field distribution is symmetrical and relatively uniform on the XZ and YZ planes, while the magnetic field distribution on the XY plane is only symmetrical about the extension of the core diagonal. According to Faraday’s law of electromagnetic induction and the right-hand spiral law, in the corner area where the magnetic field distribution is the smallest in Figure 6, the magnetic fields generated by the two rectangular coils are almost opposite in direction and cancel each other out.

The magnetic field of the cross spiral coil is uniformly distributed along a diagonal line of the square magnetic core and the magnetic field strength is larger than in other areas based on the above characteristics. A new coupling structure in which four cross coils are superimposed on a plane is designed, as shown in Figure 7.

The magnetic field distribution of this coil is deduced by Maxwell’s equations and Bio–Savart (Equation Figure 5 and Equation Figure 6) [37,38,39,40,41,42].
(6)$$dB=\frac{\mu_{0}}{4\pi }\frac{Idlsin\theta }{r^{2}}$$
(7)$$\left\{ \begin{array}{l} \oint_{l}\vec{H}d\vec{l}=\int_{s}\gamma \vec{E}d\vec{S}+\int_{S}\rho \vec{v}d\vec{S}+\int_{s}\frac{\partial \vec{D}}{\partial t}d\vec{S}\\ \oint_{l}\vec{E}d\vec{l}=-\int_{s}\frac{\partial \vec{B}}{\partial t}d\vec{S}\\ \oint_{S}\vec{B}d\vec{S}=0\\ \oint_{S}\vec{D}d\vec{S}=0 \end{array}\right.$$

The first and second formulas of Maxwell’s Equation (7) describe the relationship between electric and magnetic fields, which can be converted into each other. The third formula states that the magnetic flux through any closed surface is equal to zero, and the magnetic field is passive. The fourth formula describes the relationship between the electric flux through any closed surface and the charge within that closed surface. In the process of mutual conversion between the electric field and the magnetic field, the direction of the magnetic field generated by the electric field can be determined by Ampere’s law, and the direction of the induced electromotive force can be determined by Lenz’s law.

Combined with Bio–Savart’s Law, Maxwell’s equations, and Lenz’s law, the direction of the current in each coil is adjusted so that the magnetic field generated by each coil is directed from the coil to the center. The superposition results in a dense, uniform, and centrosymmetric magnetic field. The schematic diagram of the derivation process is shown in Figure 8.

The cross helical coil structure is a mutually superimposed enhancement area, while the magnetic field intensity on the other diagonal is relatively small. The uniformity and symmetry of the magnetic field generated by the overall coil are not enough. The new coil moves the coil originally located in the center to the outside of the magnetic core, retains the diagonally enhanced area of the original cross coil, removes the area with the weak magnetic field, and obtains a larger magnetic field enhancement range. Four identical coils and four corners are placed symmetrically about the center to make the magnetic field center-symmetrical. Finally, the quadrilateral overlapping coupled coil structure is obtained, and the current flow direction, plane magnetic field distribution, and coupling mode on each coil section are shown in Figure 9.

The simulation results are consistent with the plane magnetic field distribution theoretically analyzed by the quadrangular overlapping coupled coil structure. At the same time, the upper and lower coils of the space structure are symmetrical about the center of the z-axis, and the magnetic field generated by the coils is also symmetrical about the center of the z-axis. Compared with the previous coil structure, the magnetic field distribution in the central area of the quadrilateral overlapping coupling coil structure is more uniform and concentrated. In theory, when the coil is offset, the change rate of the mutual inductance is relatively smaller.

#### 3.3.2. Simulation of the Anti-Offset Capability of the New Coil

The coupling structure parameters of the simulation model are shown in Table 1.

Define the change rate of the mutual inductance coefficient by dividing the mutual inductance coefficient after an offset by the mutual inductance coefficient when there is no offset. Figure 10 shows the variation curve of mutual inductance obtained after the three kinds of coil structures are shifted by a certain distance in the horizontal direction.

When each structure is offset by 80 mm, the mutual inductance of the new structure can be maintained at 75.06% of the starting value, while the cross coil and circular coil are 71.63% and 31.86%, respectively. It is not difficult to see that when the coil is offset, the change rate of the mutual inductance of the new structure is significantly lower than that of the circular coil, and is more stable than that of the crossed coil. When the offset is small, the change of the mutual inductance of the new structure is very small, and the offset can be kept within 95% within 30 mm. When the offset is large, the mutual inductance can remain stable. In summary, the abovementioned new coils have strong radial offset resistance and have the potential to ensure that the energy transmission efficiency is always stable and efficient during the energy transmission process.

#### 3.3.3. Structural Adjustment and Optimization of the New Coil

The thickness of the iron core and the presence or absence of the core can affect the deflection resistance of the structure. The changes in mutual inductance when the new coil is facing and offset under different core thicknesses are shown in Figure 11 and Figure 12.

As shown in Figure 11, when the coil is offset by 100 mm, the mutual inductance of the structure with a magnetic core can be maintained at about 62%, while the mutual inductance of the structure without a magnetic core is reduced to 48.2% at the beginning. With the change of the coil offset distance, the change rate of the mutual inductance of the coil without iron core is larger than that of other coils with iron core. The variation of mutual inductance obtained by iron cores with different thicknesses is basically the same.

As shown in Figure 12, the larger the core thickness value, the greater the positive mutual inductance. The mutual inductance coefficient obtained by the coreless simulation is 13.2394 MH, and when the iron core thickness is 8 mm, the mutual inductance coefficient reaches 630.014 MH. It is concluded that the iron core in the coil is an indispensable part. Its existence is beneficial to the anti-offset performance of the coil, but the thickness of the iron core has little effect on the performance, and thereby the thickness of the coil can be flexibly adjusted according to the actual situation.

## 4. Experimental Verification

### 4.1. Construction of the Experimental Platform

To test the characteristics of the anti-offset capability of the new quadrilateral overlapping wireless energy transmission coupling coil structure, a wireless energy transmission system platform, as shown in Figure 13, was built.

The platform consists of a simple topology and novel coils. Select the core type according to the simulation environment: core material ferrite core. The experimental platform mainly includes a transmitting coil power supply device, a new type of magnetic coupling coil mechanism (consisting of a transmitting end coupling structure and a receiving end coupling structure, as shown in Figure 13), a receiving coil power supply device, a DC resistance load, etc. The transmission efficiency is obtained by testing with an oscilloscope. The voltage on the input side is directly derived from the voltage source, and the current is measured by the oscilloscope. There is a resistive load on the output side, and the power on the output side can also be obtained by measuring the current flowing through the load with the oscilloscope.

The new quadrilateral overlapping cross-coupling coil is composed of four independent improved cross-coupling coils which are overlapped and wound according to the size designed by the simulation. Each independent cross coupling structure adopts an iron core with a length and width of 10 × 10 cm and a thickness of 2.5 mm. Each independent winding is wound with 18 turns, and the solenoid is wound on both sides of the outer side of the magnetic core and fixed. Finally, a new type of quadrilateral overlapping coil with length and width of 20.5 cm is formed. The current in each winding is connected according to the simulation analysis.

To verify that the new coupling coil structure proposed in this paper has a higher coupling coefficient and better anti-offset performance for wireless charging systems, the power changes under the influence of different gaps, different loads, and different structures are subsequently analyzed.

### 4.2. Experiment on the Anti-Offset Capability of the New Coil

First, the change of energy transfer efficiency when the new coil is offset is tested, and the resistance to radial (lateral) offset is tested by comparing it with circular coils of similar size and the same number of turns. Keep the power supply voltage of 20.98 V, the load resistance of 15 Ω, and the coil interval of 10 cm unchanged; the initial position of the coil is positive, and the receiving end is moved radially by 1 cm each time to test and record the transmission power of the new coil and the circular coil, respectively. The specific values of transmission power when coils face each other are shown in Table 2 below, and the offset results are shown in Figure 14:

It can be seen that when the coil model and specifications are tested strictly according to the simulation parameters, the experimental test results are in good agreement with the simulation theory. When each offset is 90 mm, the transmission efficiency of the circular coil is only 30.03%, while the transmission efficiency of the new structure is 67.42%. Compared with the circular coil, the anti-deflection ability is significantly improved. The new coil structure has good anti-offset performance.

Then, the stability of the energy transfer efficiency and the anti-offset characteristics of the new coil under different loads and different distances (axial offset) were tested. The coil spacing remains 10 cm when the load changes and the load resistance remains 10 Ω when the spacing changes. The energy transfer efficiency is the ratio of the transmission power to the power supply. The experimental results are shown in Figure 15 and Figure 16.

It can be seen from the above experimental results that when the load resistance changes, the transmission efficiency of the wireless energy transfer system using the new coil is still relatively considerable and stable, the effect of resisting offset is also good, and the difference between the curved edges under different loads is small; when the coil spacing changes, the transmission efficiency changes greatly, but the trends of different curves are less different. It is not difficult to see that the new structure has good anti-migration performance under different loads and different spacings.

## 5. Conclusions

This paper starts from the problem that the offset of the coupling coil affects the transmission efficiency in the wireless power transmission system, aiming at the anti-offset method of improving the coil structure to homogenize the magnetic field. The existing anti-offset coils were studied and analyzed, and a new anti-offset coil was innovatively designed. The quadrilateral overlapping wireless power transmission coupling coil structure was simulated and experimentally verified for its energy transfer characteristics and anti-offset ability. It was further verified by simulation that it has excellent performance since the energy mutual inductance coefficient does not change much when migrating within a certain range. A magnetic coupling resonant wireless power transmission test platform was built, and tests for the comparison experiment of the anti-excursion ability of the new coil and the circular coil, the anti-excursion ability experiment under different loads or spacings, and the anti-excursion ability of the small circular coil as the receiving end were carried out. In the previous simulation and experiment, by comparing other coil structures, the corresponding experimental parameters were measured. It was verified that the new coil has excellent anti-offset characteristics. The quadrilateral overlapped wireless power transmission coupling coil has certain application prospects in scenarios such as the wireless charging stations of electric vehicles and drones.

## Figures and Tables

**Figure 1 sensors-22-05955-f001:**
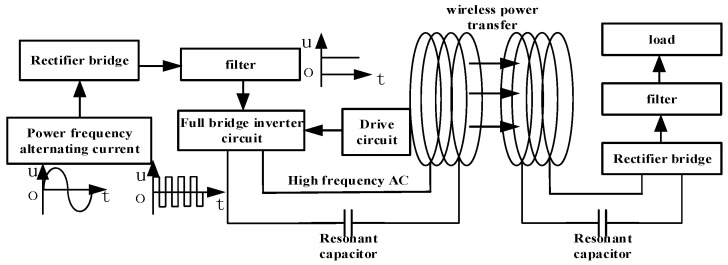
Basic schematic of radio charging.

**Figure 2 sensors-22-05955-f002:**
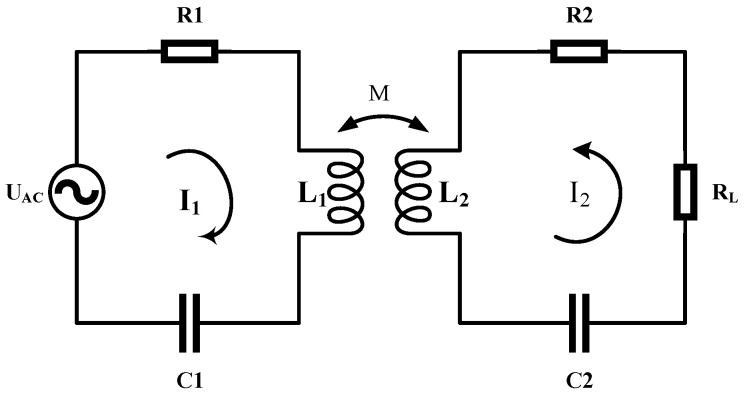
Magnetically coupled resonant circuit model.

**Figure 3 sensors-22-05955-f003:**
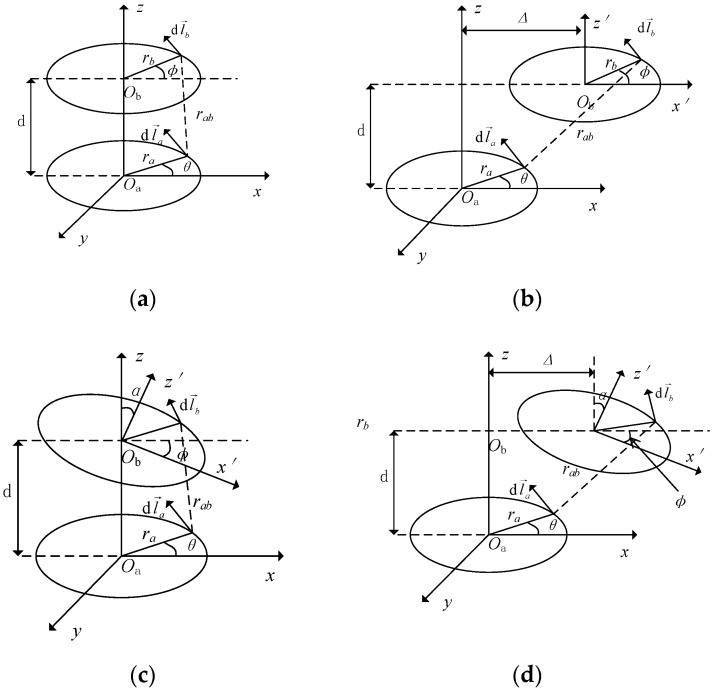
Schematic diagram of coil offset: (**a**) ideal alignment; (**b**) radial offset; (**c**) angular offset; (**d**) angular and radial offset.

**Figure 4 sensors-22-05955-f004:**
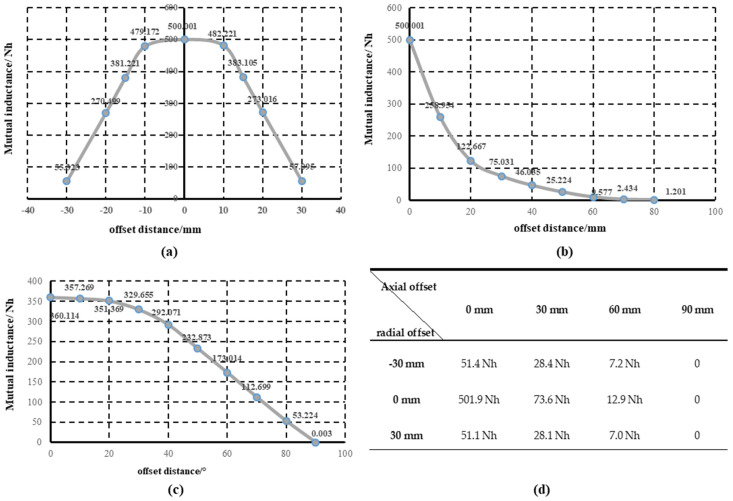
Mutual inductance changes when the coil offsets: (**a**) effect of radial offset on mutual inductance; (**b**) influence of axial offset on mutual inductance; (**c**) influence of angle offset on mutual inductance; (**d**) simultasneous radial and axial offset changes in mutual inductance.

**Figure 5 sensors-22-05955-f005:**
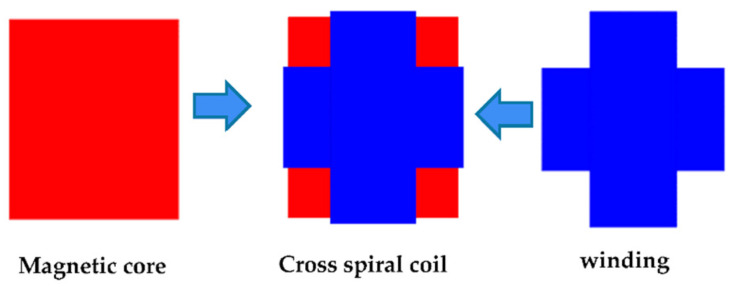
Schematic diagram of the cross-coupled coil structure.

**Figure 6 sensors-22-05955-f006:**
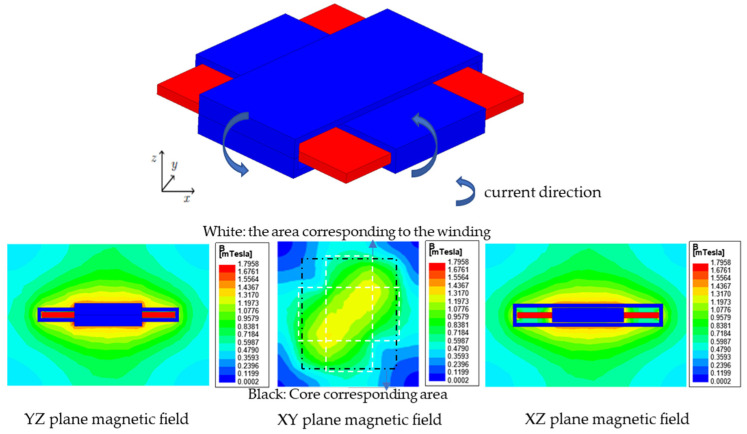
Magnetic field distribution of cross coupling coils.

**Figure 7 sensors-22-05955-f007:**
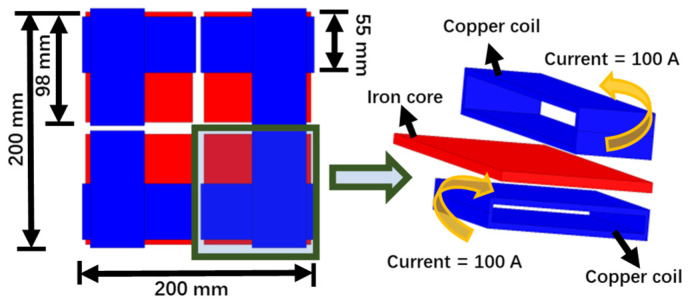
Schematic diagram of the new coil structure.

**Figure 8 sensors-22-05955-f008:**
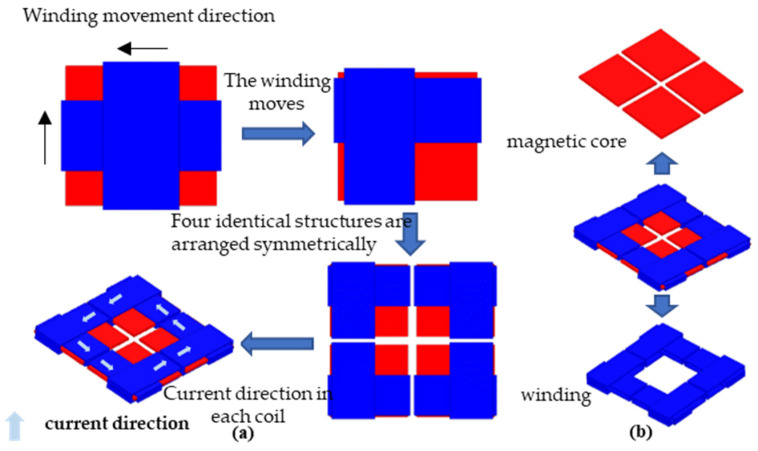
Derivation of the new coil structure: (**a**) coil structure derivation and current direction; (**b**) coil structure composition.

**Figure 9 sensors-22-05955-f009:**
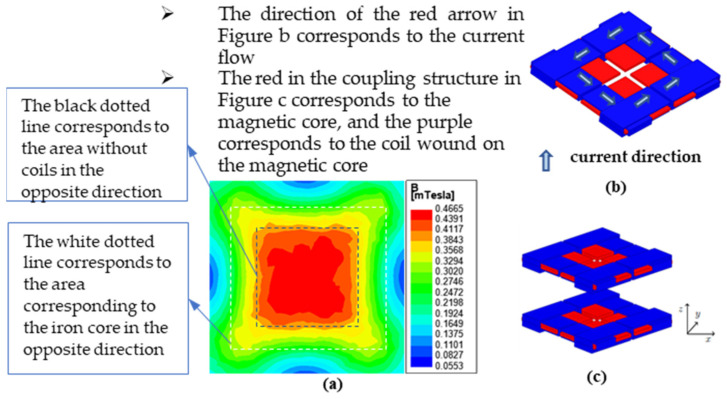
Magnetic field distribution, current flow, and coupling method: (**a**) XY plane magnetic field distribution; (**b**) XY plane magnetic field distribution; (**c**) schematic diagram of coupling structure space.

**Figure 10 sensors-22-05955-f010:**
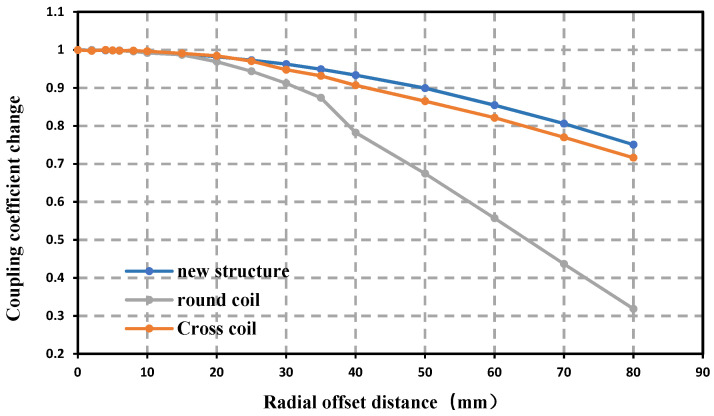
Mutual inductance changes of three coils during offset.

**Figure 11 sensors-22-05955-f011:**
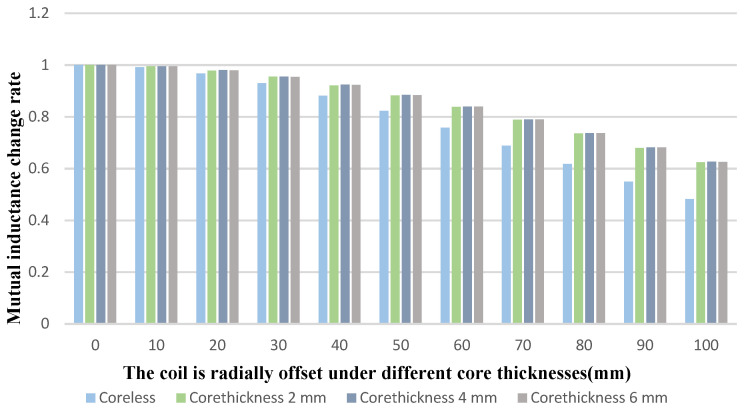
Variation curve of mutual inductance during offset under different core thicknesses.

**Figure 12 sensors-22-05955-f012:**
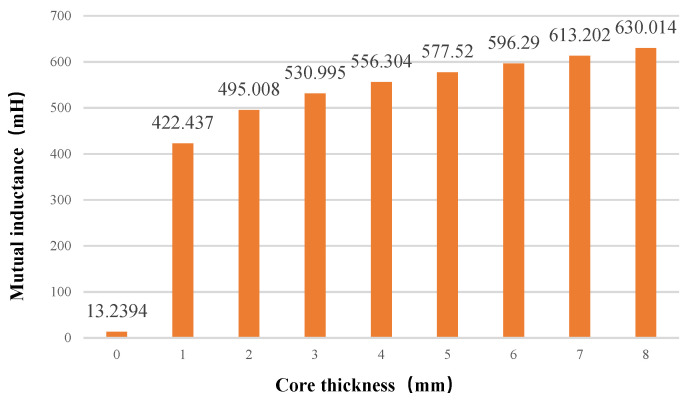
Relationship between mutual inductance and core thickness when facing each other.

**Figure 13 sensors-22-05955-f013:**
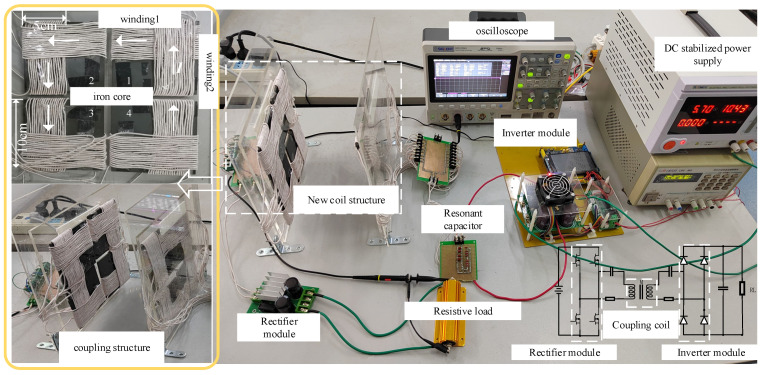
Experimental platform.

**Figure 14 sensors-22-05955-f014:**
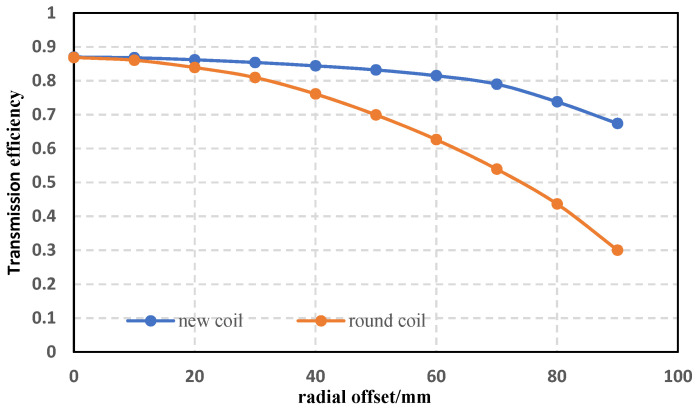
The magnitude of the change in the transmitted power when the new coil and the circular coil are offset.

**Figure 15 sensors-22-05955-f015:**
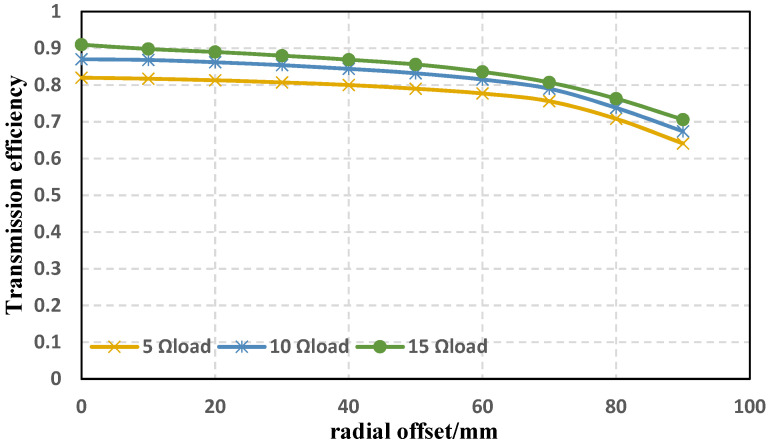
Transmission efficiency and anti-offset capability of the new coil under different loads.

**Figure 16 sensors-22-05955-f016:**
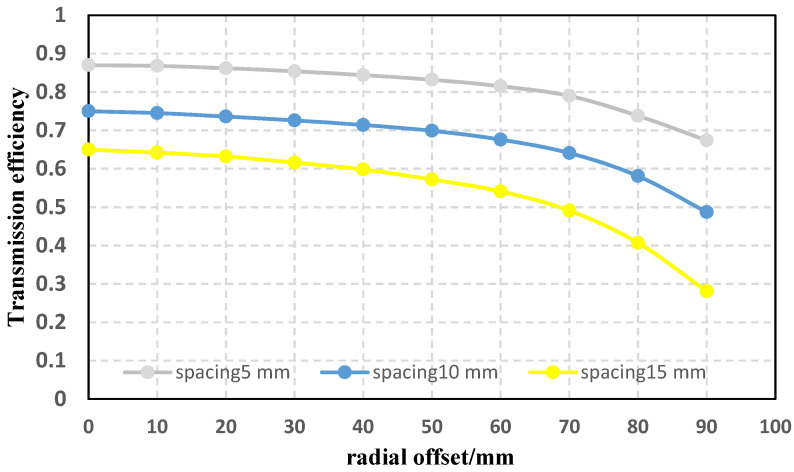
Transmission efficiency and anti-offset capability of the new coils at different spacings.

**Table 1 sensors-22-05955-t001:** Structural parameters of the simulated coil model.

Structure Name	Round Coil	Cross Coil	New Structure
Coupling form	** 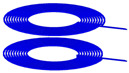 **	** 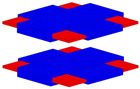 **	** 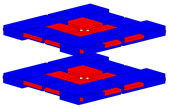 **
Coil size	** 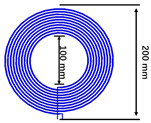 **	** 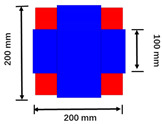 **	** 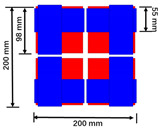 **

**Table 2 sensors-22-05955-t002:** Transmission power of each structure input and output when facing each other.

Coil Type	New Structure	Round Coil
Input Power (W)	40.924	36.907
Output Power (W)	35.604	32.057

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
