# Peer review of "Structure Design of Quadrilateral Overlapped Wireless Power Transmission Coupling Coil"

_sensors, 2022, doi:10.3390/s22165955_

Round 1
Reviewer 1 Report
The manuscript shows interesting results. I would like to make the following comments:
1. In Fig. 10, I do not see too much advantage of the new structure over the cross coil which is much simpler than the new one.
2. Fig. 11 is rather incomplete. It should be either completed or deleted.
Reviewer 2 Report
Dear authors,
Congratulations for the interesting paper. However, there are several sentences that must be rewritten in order to be understandable by readers:
-Three first lines of the abstract
-Three first lines of subsection 3.1
-First sentence inmediately below Figure 4
-Although many people know about Maxwell, Biot-Savart and Lenz equations, it is advisable that you explain a bit more the set of equations in (7). Or at least explain the content of each letter/variable.
Reviewer 3 Report
The paper presents an interesting problem dealing with wireless power transfer and probably the authors have done good work but the manuscript needs to be improved.
The manuscript must follow the template of Sensors journal.
It is not clear which kind of practical application is addressed by the authors. I suggest that the authors better highlight the application they have in mind so that the reader may understand the interest of the work.
It is also unclear how transmission efficiency was determined.
What was the transmission power of the wireless system in the case of the experiments?
"It is verified that the new coil has excellent and stable anti-offset characteristics ..." This is not clear at all, and should be extensively rewritten and explained.
The conclusions should be based on the results obtained and not general statements. You should use values to justify statements from conclusions.
Round 2
Reviewer 1 Report
Authors essentially are saying that there is no improvement with the new structure but there is room for improvement. If the situation is like this, until there is improvement, it is recommended to delete the new structure to shorten the length of the manuscript. For Fig. 11, Authors have to do something to make the figure more clear because the current figure is very confusing. If it can't be done, just simply eliminate it.
Reviewer 3 Report
The conclusions should be based on the results obtained and not general statements. You should use values to justify statements from conclusions.
